# Fragmentation Attacks and Countermeasures on 6LoWPAN Internet of Things Networks: Survey and Simulation

**DOI:** 10.3390/s22249825

**Published:** 2022-12-14

**Authors:** Sarah Alyami, Randah Alharbi, Farag Azzedin

**Affiliations:** 1Information & Computer Science Department, King Fahd University of Petroleum and Minerals, Dhahran 31261, Saudi Arabia; 2Applied College, Imam Abdulrahman bin Faisal University, Dammam 31441, Saudi Arabia; 3Interdisciplinary Research Center for Intelligent Secure Systems, King Fahd University of Petroleum and Minerals, Dhahran 31261, Saudi Arabia

**Keywords:** IoT, security, 6LoWPAN, wireless sensor networks, IPv6

## Abstract

The Internet of things is a popular concept in the current digital revolution. Nowadays, devices worldwide can be connected to the Internet, enhancing their communication, capabilities, and intelligence. Low-Power Wireless Personal Area Network (6LoWPAN) was specifically designed to build wireless networks for IoT resource-constrained devices. However, 6LoWPAN is susceptible to several security attacks. The fragmentation mechanism, in particular, is vulnerable to various attacks due to the lack of fragment authentication and verification procedures in the adaptation layer. This article provides a survey of fragmentation attacks and available countermeasures. Furthermore, the buffer reservation attack, one of the most harmful fragmentation attacks that may cause DoS, is studied and simulated in detail. A countermeasure for this attack is also implemented based on a reputation-scoring scheme. Experiments showed the harmful effects of the buffer reservation attack and the effectiveness of the implemented reputation-scoring countermeasure.

## 1. Introduction

The Internet of things (IoT) involves a network of connected devices that are able to collect and transfer data wirelessly using the Internet. IoT is applied in many environments, such as within home appliances, medical systems, and transportation networks. The IoT concept utilizes Wireless Sensor Networks (WSN), which are composed of a group of sensors used to collect and transmit data for specific tasks. IoT devices are often resource-constrained, and to be able to connect to the Internet, Low-Power Wireless Personal Area Network (6LoWPAN) is utilized. 6LoWPAN enables Internet connection to IoT networks using Internet Protocol version 6 (IPv6). 6LoWPAN employs the IEEE 802.15.4 standard, characterized by restricted memory capacity, low energy, and limited processing power [1,2,3]. 6LoWPAN introduces a new layer (adaptation layer) in the Open Systems Interconnection (OSI) model. The adaptation layer lies between the network layer and the data link layer, as shown in Figure 1. The adaptation layer is needed to solve compatibility issues (e.g., transmission range, memory, and energy constraints) between the IPv6 protocol and IEEE standard 802.15.4. The adaptation layer is also responsible for the header compression of IPv6 packets [4]. There are two categories of routing schemes, mesh-under and route-over. In mesh-under, the adaptation layer takes the routing decision, and the packet is forwarded to the next hop after all of its fragments are received and reassembled. In route-over, the network layer takes the forwarding decision. The packet is sent to the network layer after its fragments are received and reassembled. The network layer will send the packet back to the adaptation layer if the packet needs to be forwarded [5]. There are no mesh-under protocols specified by the Internet Engineering Task Force (IETF) [6]. One route-over protocol is recommended by the IETF, which is the “Routing Protocol for Low Power and Lossy Networks” (RPL) [6] used to route packets in WSN. End-to-end communication in 6LoWPAN/RPL networks follows a Destination-Oriented Acyclic Graph (DODAG). In DODAG, there is a gateway node (root node), the internal node for routers and IoT devices, and leaf nodes for IoT devices. Each node is assigned a rank. Leaf nodes have a higher rank than internal nodes, and the root node has a rank of zero [3,7,8].

6LoWPAN does not ensure end-to-end security; it only ensures hop-by-hop security provided by the IEEE 802.15.4 mechanism in the link layer. However, the upper layers are responsible for providing end-to-end security. Vulnerabilities of 6LoWPAN are inherited from both the Internet Protocol (IP) as well as the devices’ resource-constrained nature. One of the mechanisms to secure communication over IP is IPSec. It encrypts the payload, ensures its integrity using the encapsulating security protocol, and confirms the sender’s origin using the authentication header. However, IPSec is not suitable for networks with limited-resource devices, such as 6LoWPAN, in that it needs additional bits and resources (processing and energy) to perform its security functionalities [9].

Moreover, an essential aspect of 6LoWPAN is the fragmentation mechanism. The constraints on the lower power sensor nodes limit the frame size to around 127 bytes. Because the IPv6 packet size exceeds the IEEE 802.15.4 frames size at the link layer, packets need to be fragmented to be transmitted and reassembled at the adaption layer. This fragmentation mechanism is vulnerable to several attacks due to the absence of verification of fragment integrity and fragment origin. Fragmentation attacks exploit the packet fragmentation process in 6LoWPAN. Malicious entities may send fabricated or duplicated fragments that may overload the network. Examples of such attacks include replay attacks, duplication attacks, spoofing, and buffer reservation [3].

The security requirements for 6LoWPAN are specified by the Network Working Group in RFC 4919 [10], and include integrity, confidentiality, availability, freshness, assurance, energy efficiency, resiliency, robustness, and authentication, as seen in Figure 2.

Several measures can be taken to enhance the security in 6LoWPAN, such as cryptography [11], Intrusion Detection Systems (IDS) [12,13] and reputation-scoring (trust models) [3,14]. Recently, various techniques have been proposed to protect against these attacks, such as detecting fragment modifications in [15], fragment duplication and buffers reservation attacks in [16], and spoofing attacks in [17].

### 1.1. Motivation and Contributions

We were motivated by the recent attention paid by research groups [18,19,20,21,22] to 6LoWPAN fragmentation attacks. These researchers identified 6LoWPAN fragmentation attacks as an open and challenging IoT security issue. Comprehensive surveys and classifications are dedicated to security attacks in IoT environments where major vulnerabilities, including 6LoWPAN vulnerabilities, are discussed and analyzed.

As such, this article aims to investigate fragmentation attacks on 6LoWPAN IoT networks. Related studies are surveyed to categorize fragmentation attacks and existing mitigation schemes. Furthermore, a simulation of a buffer reservation attack is carried out to analyze and evaluate the repercussions of this attack. The results shed light on the severity of such an attack. This study may benefit security analysts and IoT network engineers, as the analysis of the simulated attack will help understand the full impact of the attack. The analyzed parameters can be used as features to help swiftly detect buffer reservation attacks. Moreover, a countermeasure based on reputation scoring is also implemented and assessed in this article. Therefore, the contributions of this article can be summarized as follows:Surveys papers on 6LoWPAN fragmentation attacks and their proposed countermeasures;Highlights open issues in this domain and provides a comparative analysis of existing solutions;Simulates the buffer reservation attack and analyzes its impact on network performance;Implements and assesses the effectiveness of a reputation-scoring countermeasure.

The rest of the paper is organized as follows. A problem statement is presented in Section 1.2. A description of 6LoWPAN specifications is provided in Section 2. Vulnerabilities of the fragmentation mechanism are discussed in Section 3. Countermeasures to secure 6LoWPAN are presented in Section 4. A literature review of related studies is presented in Section 5. Performance evaluation and results analysis and discussion are provided in Section 7 and Section 7.4, respectively. Finally, Section 8 concludes the paper and envisions future directions.

### 1.2. Problem Statement

In 6LoWPANs, as individual packets are partitioned into smaller fragments, they are vulnerable to several attacks that threaten the privacy, integrity, and availability of the transported data. Therefore, a security mechanism is needed to enable the transmission of fragmented packets while protecting their privacy. This also prevents any alteration that threatens the integrity of the fragments and guarantees the network’s resistance to Denial of Service (DoS) attacks. To address this problem, several studies have proposed countermeasures to 6LoWPAN fragmentation attacks. This paper aims to study fragmentation attacks on 6LoWPAN and discusses the currently available countermeasures. Moreover, this paper presents a security analysis of the buffer reservation attack, which is considered a highly severe DoS attack. Possible attack scenarios are analyzed to investigate these attacks’ severity and feasibility. To achieve the objectives of the study, we survey all related studies in order to answer the following questions:RQ1: What is the fragmentation mechanism of 6LoWPAN?RQ2: What are the security vulnerabilities due to the 6LoWPAN fragmentation mechanism?RQ3: What are the available countermeasures to those attacks?RQ4: What is the impact of a buffer reservation attack on the performance of the affected nodes?

## 2. 6LoWPAN Specification and Fragmentation

### 2.1. Specification

6LoWPANs are networks specifically designed for wirelessly connecting devices with restricted power and memory. The specifications of 6LoWPANs are as follows: devices are of limited power (usually battery operated), devices are placed in an ad-hoc manner, a generally large number of devices, devices tend to be unreliable, networks are of lower bandwidth rates (250 kbps–20 kbps), and, most importantly, a maximum MAC layer frame size of 127 bytes. The maximum size of frames in 6LoWPAN is 127 bytes, and the IPv6 packets need a Maximum Transmission Unit (MTU) of 1280 bytes [3]. If the packet size is less than the maximum payload length of 6LoWPAN, it will be sent directly without any fragmentation [23]. Otherwise, it must be fragmented into smaller sizes if it does not fit the specified frames. The adaptation layer performs this fragmentation in 6LoWPAN [3].

### 2.2. Fragmentation

#### 2.2.1. Fragments Header

A fragmentation header is defined in each fragment and is used for reassembling fragments. All fragments contain a fragment header. The first fragment header (FRAG1) includes three fields: (1) the dispatch field, which differentiates between FRAG1/FRAGN. For inastance FRAG1 contain the dispatch 11000 and FRAGN contain the value 11100; (2) the datagram size field (11-bit), which holds the size of the whole packet used to reserve the buffer in the received node; and (3) the datagram tag field (16-bit), which holds a unique sequence number. The following fragments carry a (FRAGN) header, which contains the datagram offset field (8 bytes) to indicate the fragment position in the original packet [3]. The payload length of each fragment should be a multiple of eight [23]. Routing information with IPv6 addresses is provided in the first fragment header [3]. Figure 3 shows fragment headers. Routing is performed for the remaining fragments by using the datagram tag to map the current fragment to the previously received first fragment (FRAG1) [3,16].

#### 2.2.2. Fragments Reassembly

When the receiver receives fragments, it starts constructing the original packet [24]. The receiver identifies fragments based on three values: the link layer source and distention addresses, the datagram size, and the datagram tag [23]. The receiver uses the datagram offset to place the packet on its appropriate location in the reassembly buffer [23,24]. The receiver reserves space on the buffer with the value stated in the datagram size field. If the receiver receives a fragment that overlaps with currently buffered fragments, it should discard the saved fragment and keep the fresher one. Moreover, the receiver should start a timer upon the packet’s reassembly; if all fragments are not received before the time-out, all fragments must be discarded [24]. The memory requirement for each reassembly buffer entry is at least 1302 bytes, which should be reserved by the node for the reassembly [23].

#### 2.2.3. Forwarding Fragments

Fragment forwarding is guided by the IPv6 address mentioned in the FRAG1 [23]. Typically, the IPv6 packet is fragmented and reassembled at each hop of the network [25]. Consequently, enough memory to store all fragments should be provided in each hop, and the time needed to receive all fragments bounds the packet delivery time. Moreover, some nodes in the network are connected to more nodes than others, and they will need more memory than others to manage the load of their downstream nodes’ reassembly. The memory requirement for packet reassembly is considered to be high for constrained devices; hence, some studies have proposed other fragment-forwarding techniques [23].

## 3. Fragmentation Security Vulnerabilities

The fragmentation procedure in the adaptation layer suffers several vulnerabilities due to poor fragment authentication and verification mechanisms. Fragmentation attacks disrupt the process of reassembling fragmented packets at the destination. To describe potential attacks, we present a scenario similar to that analyzed by Hummen et al. in [16]. The scenario illustrated in Figure 4 depicts three malicious attackers: an external attacker (denoted as E), a forwarding internal attacker (denoted as F), and a non-forwarding attacker (denoted as N). F and N are attackers internal to the 6LoWPAN, and E is outside the network. Resources are denoted as (R), and a gateway (denoted as GW) is used to connect the 6LoWPAN to the Internet. The arrows show the direction of some defined forwarding routes. Considering the attackers’ location within the network, we assume E (external entity) is equipped with more resources than the limited-resource nodes within the 6LoWPAN. Therefore, a possible threat from E is to flood the network with packets that overwhelm the resource-limited 6LoWPAN nodes. As for N, which is located near a forwarding path, as seen in Figure 4, it can eavesdrop from the nearby channel and issue attacks based on the overheard data. On the other hand, F is positioned within the forwarding path, enabling it to discard, alter, or duplicate desired fragments maliciously. Flooding attacks from outsider attackers may be prevented by enabling rate-limiting by the gateway, which prevents responding to the flooding messages [3]. Given the previously mentioned scenario, possible network-internal attacks are described in the following subsections.

### 3.1. Fragment Duplication

Duplication attacks exploit the inability of a node to differentiate between legitimate fragments and maliciously duplicated ones, which may happen when an internal attacker located near the forwarding channel (N) overhears the transmitted data and sends a duplicate fragment to the same node. Furthermore, an attacker located within the forwarding path (F) can receive a fragment, duplicate it, and send it to the next node in the forwarding path, as seen in Figure 5. Moreover, the external attacker (E), which has more power and resources than the receiver node, can amplify this attack by sending multiple large packets; the gateway will fragment these packets and the receiver should process more packets [26]. As a result, the receiving node cannot differentiate between legitimate and duplicated fragments [27]. Therefore, the receiving node cannot deal with the duplicated fragments when attempting to reassemble the fragmented packets, i.e., it will not know which fragment to select for the reassembly [26]. In this situation, the victim node may drop the packet, as it appears corrupted. Therefore, the packet will need to be re-sent. This vulnerability opens the space for a Denial of Service (DoS) attack [27]. As this attack continues, the 6LoWPAN resources will ultimately be exhausted.

### 3.2. Fragment Alteration

Alteration attacks exploit the lack of fragment integrity verification in the adaptation layer in 6LoWPAN. An insider attacker who is part of the forwarding path (F) receives the fragment, alters the payload, and forwards it to the next node, as seen in Figure 6. The receiving end cannot detect the forged fragment [3]. This attack violates the integrity and availability of the data, as the transmitted data may be altered or damaged, and prevents the reassembly of the packet fragments.

### 3.3. Spoofing

Spoofing attacks are based on an attacker impersonating another entity by binding its IP address with the victim’s MAC address or its own MAC address to the victim’s IP address [17]. They may be harmful in several ways. For instance, an insider attacker (N or F) impersonates another node (e.g., Node 1) and issues fragments with Node 1’s IP address as the source ID within the FRAG1 header. The attacker sends the forged fragments to another node, which believes that the fragments’ source is, in fact, Node 1. The forged fragments may contain false or harmful data that harm the target node. Another scenario is when an attacker binds its IP address with Node 1’s MAC address and directs all the incoming traffic away from Node 1 [28], as illustrated in Figure 7. Spoofing attacks take advantage of the lack of a sender-authentication mechanism in the 6LoWPAN. Spoofing could be used to carry out future attacks such as DoS, distributed DoS, flooding attacks, and man-in-the-middle attacks. Thus, it can prevent the victim node from creating relationships with neighbors. DoS attacks can be created by attaching the MAC address of the attacker to the victim’s IP address, causing the victim node’s inability to register itself to the router, which leads to a corrupted routing table. The spoofing attack damage will persist until the attacker node is detected [17].

### 3.4. Replay Attacks

Replay attacks aim to overwhelm the 6LoWPAN with fragments that were previously received. These attacks exploit the lack of a mechanism to check the freshness of the data in the 6LoWPAN adaptation layer [3]. In the given scenario, the insider attacker (F) receives fragments, stores them, then forwards them. Later on, F re-sends the same fragments, exploiting the fact that the receiving node will not know that the fragments were received earlier. Replay attacks are depicted in Figure 8. This attack exhausts resources with the replayed fragments, thereby delaying the transmission of legitimate fragments or causing DoS [3].

### 3.5. Buffer Reservation

As shown in Figure 9, buffer reservation attacks exploit the nodes’ limited resources in 6LoWPANs. The vulnerability exploited in this attack is that the receiving nodes need to reserve enough space within their buffer, as indicated by the packet size in the FRAG1 header. Space is used to reassemble the entire packet from the received fragments. A node cannot receive other fragments if its buffer is reserved or occupied. In 6LoWPAN, the standard time-out limit to reassemble a packet is 60 s. When the transmission period times out, the node discards the unassembled fragments. An insider attacker (F or N) sends a fragment with header FRAG1 to the victim node in this attack. The victim node reads the packet size (in the datagram field), reserves the buffer’s space accordingly, and awaits the remaining packet fragments. However, the attacker does not send the remaining fragments, and when the time-out limit is reached, the attacker again sends the FRAG1 fragment, which reserves the buffer again. If this attack continues, the victim node can never receive any legitimate fragments and is tied up attempting to process the attacker’s malicious fragments [3].

### 3.6. Impact and Severity of Attacks

Possible attacks related to packet fragmentation in 6LoWPAN are: duplication, alteration, spoofing, replaying, and buffer reservation. Alteration attacks target data integrity; duplication, spoofing, replay, and buffer reservation risk the network’s availability and are considered Denial of Service (DoS) attacks. In terms of severity, DOS-based fragmentation attacks are the most severe and can cause serious disruption of the WSN. In order to counter duplication attacks, a mechanism must be implemented that allows for the distinguishing of original fragments and duplicates, such as Message Authentication Codes (MAC). If a duplicate is detected, a policy needs to be defined that handles duplicates without the need to drop and resend the packet [15]. Alteration attacks can be detected by employing fragment integrity checks such as hash functions [29]. Replay attacks may be prevented by using nonce and timestamps to verify the fragments’ freshness [24]. Spoofing attacks can be mitigated by lightweight cryptography authentication [28] or temporary private IP addresses [17]. Finally, buffer exhaustion can be reduced by introducing new policies controlling the reservation of buffer capacity [3].

## 4. Countermeasures to Secure 6LoWPAN

This section presents some techniques used to enhance general security in 6LoWPANs, such as cryptography, IDS, and reputation-scoring systems. These mechanisms may be implemented to secure the network layer, prevent routing attacks on RPL or the 6LoWPAN adaptation layer, and mitigate fragmentation-related attacks. A description of each is provided below.

### 4.1. Cryptographic Techniques

Cryptography can be used to ensure data privacy. It can also be used for authentication purposes. However, the application of cryptography in 6LoWPAN is restricted by the limited computation and memory capacities of 6LoWPAN devices. Therefore, encryption algorithms that are lightweight but sufficiently powerful are best suited for this situation. Other issues to consider when using cryptography in WSNs are key generation and distribution amongst nodes that are frequently added or removed from the WSN, along with the ability to renew or revoke keys robustly and efficiently [30]. The process of exchanging keys between nodes can also be challenging; scalability is a major concern with key exchange models [12]. Moreover, it is worth noting that cryptography techniques may be rendered useless in cases where internal attacks maliciously obtain keys and, therefore, these attacks may go undetected [12]. Ayson et al. proposed a modified RSA and elliptic curve cryptography in [31] to secure 6LoWPANs. Using cryptographically generated addresses can protect against spoofing attacks [17]. Moreover, a cryptography countermeasure that includes digital signature and encryption can protect against duplication attacks [26].

### 4.2. Intrusion Detection Systems

IDS systems are based on analyzing the system’s state to detect any signs of suspicious behavior. If an intrusion is detected, an alarm goes off and appropriate measures may be taken accordingly. As mentioned previously, cryptography may be beneficial in protecting against outsider attackers; however, insider attackers are best detected by IDSs. IDSs in 6LoWPAN need to inspect traffic between IPv6 and IEEE 802.15.4. They must also consider the limited resources in 6LoWPAN. Issues that need to be considered are: feature selection (what features of the network are most informative and are helpful for intrusion detection?), location of IDS (where should the IDS be installed within the WSN?), and analysis algorithms (what algorithms are best capable of predicting an intrusion?). Features that might indicate an attack include changes in payload, time interval between messages, and transmission delay time. IDS approaches are divided into misuse-based approaches, anomaly-based approaches, and specification-based approaches. In misuse-based approaches, a pattern of known attacks is identified and the network is monitored to detect these patterns to generate an alarm. In the anomaly-based approaches, the network’s normal behaviors are classified and then monitored to compare and detect any suspicious activities. In specification-based approaches, manually specified program behavioral specifications are used as a basis to detect attacks [12]. Machine learning has recently been leveraged to analyze behavior patterns and intelligently detect intrusions [32]. A comprehensive review for IDS in IoT networks is provided in [33]. DIS can be used to protect against duplication attacks, fragment alteration, spoofing, and replay attack.

### 4.3. Reputation Score Systems

Reputation scoring or trust models in IoT networks are built according to the behavior nodes within the WSN. A reputation score is cumulatively computed per node based on previous interactions. Trust metrics need to be defined, in which “trust” is built between an evaluator and each node. Based on the node’s reputation, the node may be given privileges, such as extended buffer reservation size, as in [3]. A well-established trust model may significantly help enhance security in WSN and provide more reliable and efficient communications [34]. Muzammal et al. [35] discussed trust models and presented several trust metrics used to calculate a node’s trustworthiness; metrics include the number of successful transactions, resource utilization, node capacity, and energy consumption. Several studies have proposed a reputation-scoring system to enhance security in 6LoWPAN, such as [3,14,36].

### 4.4. Other Countermeasures

There are plenty of other countermeasures that do not fall under the previous categories. One of these is using a temporary node private address, which protects against spoofing attack. The attack disturbance time is reduced by periodically changing the node’s private IP. Hence the spoofed address is no longer valid and the new IP address is bonded with the victim node MAC address, enabling the victim node to register itself with the router again [17]. Another countermeasure is attaching a timestamp or nonce to the fragment header to ensure the response’s freshness and that it has not been previously received [24]. Recent efforts proposed to counter fragmentation-based attacks are described in more detail in the literature review.

## 5. Literature Review

This section reviews recent related work on fragmentation attacks on 6LoWPAN. A summary of the reviewed literature is presented in Table 1. In regards to survey and review papers, Anhtuan et al. in 2012 [12] studied vulnerabilities in 6LoWPANs, particularly in RPL routing attacks such as rank attacks and local repair attacks. The study proposed an IDS scheme to mitigate these security threats. In 2018, 6LoWPAN security issues and solutions were reviewed by Chakraborty et al. [37], who reviewed and compared several proposed fragmentation security solutions in 6LoWPAN. Recently, in 2020, a comprehensive review by Muzammal et al. [35] was undertaken, which involved the security of IoT networks in 6LoWPAN and RPL protocols. The study addressed possible attacks and categorized them into RPL WSN inherited attacks, RPL-specific attacks, 6LoWPAN authentication attacks, and 6LoWPAN fragmentation attacks. However, the study focused more on RPL routing attacks.

The earliest work to address fragmentation attacks was by Kim et al. [24]. This article provided an analysis on 6LoWPAN adaptation layer threats and proposed a scheme to protect against fragment replay attacks. The author proposed the use of timestamps with unidirectional fragmented packets and nonce with fragmented bidirectional packets. However, no implementation or simulation was provided to test the proposed scheme.

Hummen et al. [16] focused on buffer reservation and fragment duplication threats. The study proposed a content-chaining procedure and buffer-splitting scheme to mitigate these threats. The content chaining is based on hash chains used to authenticate each fragment and verify that it belongs to the same packet. The buffer-splitting scheme proposes splitting the buffer into fragment-sized slots and storing fragments belonging to multiple packets to mitigate buffer reservation attacks. When the buffer is fully occupied, a mechanism is defined to select and drop one of the buffered packets based on behavior scores computed per packet.

Based on the described buffer reservation and duplication attacks in [16], Hossain et al. [3] proposed a scheme that counters these attacks in addition to replay, alteration, duplication, and spoofing attacks. The scheme was named “SecuPAN” and is based on Message Authentication Codes (MAC) that verify the fragments’ integrity to counter these attacks. The MACs are used to detect any altered fragments. Only altered fragments are requested to be re-sent instead of re-sending all the packet fragments. Furthermore, a cryptographic procedure using Elliptic Curve Cryptography (ECC) is proposed to compute the IPv6 addresses and datagram tags to prevent spoofing attacks. As for buffer-reservation attacks, a policy was defined based on the sender’s reputation to evade such attacks and manage the limited-size buffer accordingly. SecuPAN was tested on a Contiki system and showed good performance. The disadvantage of the SecuPAN scheme is the additional processing time compared to the base Contiki, as it requires nearly double the time to verify and reassemble the packet at the receiver’s end.

Buffer reservation attacks were also addressed in [14]. A reputation-based scheme named “BRAIN” was proposed to mitigate buffer exhaustion. In BRAIN, points are rewarded to nodes that successfully send consecutive fragments, which are therefore allowed to reserve larger portions of the buffer. On the other hand, nodes that only send FRAG1s and do not send the remaining fragments are deducted “trust points” and are allocated smaller portions of the buffer size. The security scheme was implemented using the Cooja simulator. In comparison to SecuPAN [3], BRAIN showed better performance in terms of throughput and packet delivery rate. However, attackers may adopt changing behaviors (on/off attack) where some packets are sent successfully and others are partially sent to reserve the buffer maliciously. This on/off behavior may elude the proposed countermeasure.

Ray et al. [38] pointed out that in previous studies, the split buffer [16], SecuPAN [3], and BRAIN [14] schemes lacked the ability to precisely identify which nodes were carrying out buffer reservation attacks. Therefore, their study proposed a buffer reservation countermeasure named “ArsPAN”, which guarantees the attacker’s identification in 6LoWPANs. The proposed scheme is an intrusion detection system based on Discrete Event Systems (DES) in the Finite State Machines (FSM) framework. This scheme involves a reputation score measure that reflects each node’s behavior and a probing packet used to disclose the attacker. ArsPAN was implemented on Contiki and showed better performance than the previous works.

Raoof et al. [36] emphasized the lack of authentication of the fragment sender in 6LoWPANs. The study proposed to use a previously proposed RPL secure mode called Chained Secure Mode (CSM) [39]. The authors suggested integrating this mode with 6LoWPAN. In this method, Network Coding (NC) creates a chain of trust along the path of routed fragments. A trust value is recorded for each neighbor. The proposed framework mitigates buffer reservation attacks by allowing only trusted fragments to be loaded into the fragment assembly buffer. A simulation was done using Cooja in Contiki OS to imitate a buffer reservation attack. The proposed CSM method demonstrated the ability to prevent the attack with a 100% packet delivery ratio. However, the authors explained that an attacker could elude the mitigation mechanism if the attacker used the victim’s link-layer address instead of the IPv6 address.

Duplication attacks were addressed in [15,26]. Hassan et al. [15] proposed a scheme based on the non-causal hash function to protect against fragment duplication attacks in the 6LoWPAN adaptation layer. The improved scheme protects from “smart” attackers that exploit causal hash schemes and modify the data without changing the hash value. Authentication of a fragment is guaranteed, and any modifications on fragments are detected. The scheme also confirms whether a fragment is, in fact, part of the same transmitted packet. However, a detailed explanation of the proposed scheme was not provided.

Nikravan et al. [26] proposed a lightweight signcryption mechanism to protect against duplication attacks. This scheme ensures integrity, confidentiality, authentication, and non-repudiation. Their scheme is based on the concept of signcryption, in which public-key encryption and digital signature operation are performed at the same time. The only drawback of this approach is that it creates a size overhead of 120% the full packet size.

As for alteration attacks, Herrero [11] proposed a dynamic Forward Error Correction (FEC) scheme to detect and correct damaged fragments. This scheme can protect against fragment alteration attacks. Since FEC is computationally expensive and battery draining, the paper proposed a method where FEC is dynamically enabled or disabled according to several parameters regarding the network’s current state. The parameters are fed into a Markov model, which predicts if FEC should be applied.

Spoofing in 6LoWPAN has also been addressed in the literature. In 2017, Mavani et al. [28] conducted a feasibility analysis for spoofing attacks on the 6LoWPANs. They created an attacker tree model to perform spoofing statistical analysis. In the attacker tree, the attack’s goal is the tree’s root and the leaves are the atomic attacks. The evaluation is performed bottom-up. They identify two paths for false IP-MAC binding. The first path is when the attacker uses spoofed 6LoWPAN-ND messages (containing the victim node MAC address and the attacker node IP address) sent to the router, causing the victim node to deregister from the router. The other path is when using a spoofed RPL message (contains the attacker node MAC address and victim node IP address) sent to the router, causing all victim node traffic to be directed to the attacker. The spoofing attack was simulated using the Cooja simulator on the Contiki operating system, and a real experiment was also conducted on the same operating system. Moreover, the study analyzed memory and power consumption and found that the energy consumption of the attacker is highly dependent on the distance between the attacker and the victim, and that the attacker code can run under the memory constraint of the low-power device of the attacker [28]. However, no measures to counter the analyzed spoofing attack were proposed in this paper. The authors continued their work in [17]. This study targeted reducing the attack disturbance window in a 6LoWPAN. They used a private addressing scheme proposed in [40,41]. They also enhanced the attack tree proposed in [28] by including temporal characteristics and, thus, adding additional parameters to the attack tree. The proposed scheme attempts to reduce the attack window by dynamically changing the node’s private IP address. However, frequently changing the IP address causes increased communication overhead. The proposed spoofing countermeasure was evaluated using the Cooja simulator and Contiki operating system. They found that the optimum value for periodically changing the address is between 60 to 90 s.

In [18], 6LoWPAN fragmentation attacks have been identified as an open and challenging IoT security issue. This article provides a comprehensive classification of the major security issues based on IoT architecture, attack implications, and application areas. In addition [20], provides a taxonomy of security attacks in IoT environments where major vulnerabilities, including 6LoWPAN vulnerabilities, are discussed and analyzed [19].

Low-level security concerns are surveyed in [22] including node sleep deprivation for power-constrained IoT nodes on the Internet of Things that are often attacked. Therefore, sensor nodes stay awake and, consequently, 6LoWPAN environments need to be customized to prevent battery drain. Furthermore, the authors of [21] demonstrated three mechanisms to improve end-to-end reliability in 6LoWPANs. Conducted performance analysis shows that the three mechanisms come with different performance and costs, making them more suitable for worse network conditions.

### Open Issues

As revealed in the literature review, several efforts have been made to address fragmentation attacks in 6LoWPAN. Table 2 provides a comparative analysis of the proposed countermeasures. In addition, a taxonomy of the attacks and proposed countermeasures is presented in Figure 10.

Replay attacks can be mitigated by introducing a nonce value and timestamps to fragments. MACs can also verify the sender and detect replay attacks [3]. However, the nonce-based scheme was not implemented and evaluated in [24], and the computation and verification MACs for each transmitted fragment can overwhelm the network.

As for alteration attacks, the proposed cryptographic non-causal hash function [15] and MACs [3] also detect alteration attacks, but with the additional overhead, as explained earlier. In addition, the forward error correction countermeasure in [11] requires redundancy bits that cause overhead to the network bandwidth.

Spoofing attacks are defended against by encrypting the IPv6 addresses in [3]. However, the need to decrypt the address for every transmitted fragment would significantly slow down the data transmission process. The temporary private IPv6 addresses proposed in [17] require additional effort to manage the periodically changed private addresses.

Duplication attacks can be prevented by content chaining [16], MACs [3], and non-causal hashing [15]. Unfortunately, the content-chaining mechanism suffers from a significant setback. It assumes that all fragments are sent in order, which may not always be accurate. Furthermore, if even one fragment is lost, the entire received packet cannot be verified by the hashing chain and is therefore dropped.

Buffer reservation attacks were mitigated by a buffer splitting scheme in [16], reputation scoring in [3,14], and IDS based on reputation scores and probing packets in [38]. The split buffer adds communication overhead, specifically by the defined packet-dropping strategy. The reputation-scoring-based countermeasures suffer from extra computation costs to keep track of the nodes’ transmission behavior. In addition, these systems may be eluded by attackers intelligently shifting their behavior to gain reputation scores and on/off strike attacks.

Buffer reservation attacks cause significant harm to the WSN and directly affect the target node’s availability, preventing it from receiving any fragments from the other legitimate nodes and leading to a DoS [3,14,38]. Moreover, the conducted literature review revealed that buffer reservation attacks are one of the most active research problems amongst fragmentation-based attacks in the past years [3,14,16,36,38]. Most papers have proposed reputation-scoring schemes to counter this attack, as seen in [3,14,36,38].

For the aforementioned reasons, a buffer reservation attack is simulated and analyzed in this study. Moreover, a reputation-scoring countermeasure is implemented and assessed. The analysis describes the impact of the buffer reservation attack on the network and the efficiency of the reputation-scoring scheme in countering this attack. The conducted simulations are described in the next section.

## 6. Proposed Reputation-Scoring Countermeasure

The implemented countermeasure is a reputation-scoring scheme, as seen in [3,14,38]. The scoring scheme calculates a score for each node based on the node’s behavior in sending packets. The model contains three behavior indicators. Each node in the WSN had a table to maintain these behavior indicators for all other nodes.

First, a reputation score (rep_score) indicator is used to reflect each node’s behavior as it transmits fragmented packets. The score is manipulated at three points:When a receiving node successfully receives and reassembles all the packet fragments, it will increment the score of the sending node by 1;If the receiving node did not receive the complete fragments of the packet and the reassembly timeout is reached, the score of the sending node is decreased by 1;If the receiver node receives fragments other than FRAG1, the sender node score will be incremented by one to give a chance to nodes that accidentally lose subsequent fragments due to a lossy connection.

The second indicator is the timeout flag, which indicates if the sending node caused the receiver to time out while waiting to reassemble the incomplete packet. Thus, the receiving node sets the timeout flag of the sender to 0. The timeout flag will be set back to 1 if the sender sends a FRAGN fragment next time. Hence, the receiving node uses the flag to determine if the sender has caused a timeout consecutively.

The third indicator is a counter (count_fail) that counts how many times the sender consecutively caused the receiver to timeout during packet reassembly. The timeout flag determines if the sender did not send the complete packet consecutive times. To counter the sender’s malicious behavior, the receiving node “blocks” the sender if it exceeds a certain threshold of consecutive timeout. The receiving node checks if the sender’s reputation score is below a certain threshold (1 in this simulation) and if the sender’s count_fail counter value is at least five, i.e., the sending node sent incomplete packets five consecutive times. If these two conditions are met, the receiver flags the sender as “blocked”.

Therefore, the receiver first checks if the sender belongs to its blocked-nodes list upon receiving fragments. The receiving node does not proceed with the sender request if the sender is in the blocked-node list.

This scoring scheme forgives a node with a high reputation score even if it has sent five consecutive incomplete packets. One drawback of the scheme is that blocking a malicious node blocks any other node that uses the malicious node as a bridge to reach the destination indirectly. Another drawback is that the attacker may change its behaviors to prevent itself from being blocked by sending normal packets frequently between malicious packets.

## 7. Simulation of Buffer Reservation Attack and Countermeasure

In this section, a simulation of a buffer reservation attack is performed to assess its impact on the WSN. Furthermore, we implemented a version of a reputation-scoring countermeasure as proposed in the literature [3,14,38].

### 7.1. Experimental Setup

An image of a Contiki 3.0 was downloaded and run through VMware and VirtualBox to conduct this study. The Cooja simulator available from Contiki was used to simulate the attack. Three scenarios are examined in this paper. The first scenario is the normal network flow without the buffer reservation attack; the second is the network flow when the attack is executed. Finally, the implemented reputation-scoring countermeasure in the third scenario is assessed as the network experiences a buffer reservation attack. We follow the same setup in [14] with a few changes, as shown in Table 3. The network topology is illustrated in Figure 11, where Node 1 is a server node, and the remaining nodes are clients. With few changes, the default “udp-server” and “udp-client” firmware were used. In this firmware, the clients send the server a message periodically (the default is every 60 s). However, we change the interval at which Node 3 sends messages. In the first scenario (no-attack), all nodes send messages every 60 s in the first trial. In the second trial, we decrease the interval to 40 s (the remaining nodes send every 60 s). The packet interval for Node 4 is set to 20 s and 10 s in the third and the fourth trials, respectively. The packet interval is changed as described above for all three scenarios (no-attack, buffer reservation attack, countermeasure with buffer reservation attack) to analyze the impact of the packet interval on the WSN. Node 3 is the designated attacker in the attack scenario and Node 1 is the victim node. Moreover, as seen in the network topology, Node 2 is out of the server’s range and would require an extra hop through Node 3 to reach the server. In order to enforce packet fragmentation, the message length is increased from the default value and the message buffer size in nodes is changed from 30 to 160 to accommodate the sent message. Fragmentation by default is enabled in this Contiki configuration, as specified by the setting SICSLOWPAN_CONF_FRAG 1 in the Contiki-default-config.h file. All Contiki default settings were used unless otherwise specified. Eclipse was installed on the image machine and connected to Contiki to edit and trace needed files.

### 7.2. Simulation of Buffer Reservation Attack

In order to implement the buffer reservation attack, changes were made to the sicslowpan.c file, which is responsible for packet fragmentation. In the attack scenario, Node 3 is the designated attacker, which periodically sends the server FRAG1 packets and does not send the remaining FRAGN packets. Upon receiving the FRAG1 packet, the server waits for the remaining packets until the timeout period is reached. The default timeout value is 20 s. The server then drops the previously received FRAG1 and resumes operating as usual. Additionally, seeing that Node 2 needs to send its messages through the attacker (Node 2 is out of server range), the attacker maliciously only forwards Node 2’s FRAG1 to the server, which prevents the complete packets from reaching the server. The “sys/node-id.h” header file is included in the sicslowpan.c file to specify the attacker’s node and its designated code in the sicslowpan.c file. Debug messages were enabled to show and trace the attack within Cooja log listener. Each simulation trial was run for 15 min. Figure 12 shows mote output describing how the packet is fragmented in a normal node. The total packet length is 199, including the header, which exceeds the maximum length specified in IEEE 802.15.4. The needed number of fragments is estimated according to the data’s size and the additional FRAGN header. The FRAG1 packet is sent first; then, the remaining two FARGN packets are sent. However, during the buffer reservation attack, the FRAG1 packet is only sent by Node 3 to the server (Node 1), as seen in Figure 13. The server times out after 20 s and drops the incomplete packet, as revealed by the mote output logs.

Figure 14 shows the server node blocking the attacker (Node 3) after its reputation score drops below 1 and sends incomplete packets more than five consecutive times.

### 7.3. Evaluation Metrics

In order to analyze the effect of the attack on the network, four evaluation metrics were used, as defined below. The evaluation metrics were implemented in the simulation script editor.

Throughput: The throughput (*TP*) is the number of transmitted packets multiplied by the total size of the packet in bits over the total simulation time, measured in bits per second (bps).
(1)$$TP=\frac{PacketsRecv\times PacketLen\times 8\times 1000}{Simulation\;Time}$$Average Energy Consumption: The energy consumption (*EC*) is computed by recording the total energy needed to transmit all the packets, which is recorded using the Contiki energest.h library. The energy consumption is calculated by multiplying the voltage and the current together with the duration that the sensor node spends operating in each of the four states, CPU, Low Power Mode (LPM), Transition (Rx), and Receive (Tx) [42]. The voltage and current are taken from the Tmote Sky sensor datasheet (https://insense.cs.st-andrews.ac.uk/files/2013/04/tmote-sky-datasheet.pdf, accessed on 16 April 2021). The Sky Mote power specifications are displayed in Table 4. In general, the energest times are in rtimer ticks, which need to be divided by Contiki’s RTIMER_ARCH_SECOND to get seconds. The energy consumption for each state is computed using Equation (Equation 2) [43]. To compute the total energy consumption of each node, the energy consumed by each state is summed. This value is averaged by the number of nodes to find the average energy consumption for the WSN.
(2)$$EC=\frac{EnergiestValue\times Current\times Voltage}{RTIMER\_ARCH\_SECOND}$$Packet Drop Ratio: The packet drop ratio (*PDR*) is the number of packets dropped over the total number of sent packets.
(3)$$PDR=\frac{totalPacketsDropped\times 100}{totalPacketsSent}$$Packet Delivery Ratio: The packet delivery ratio (*PDVR*) is the ratio of the number of received packets to the number of sent packets.
(4)$$PDVR=\frac{totalPacketsReceived\times 100}{totalPacketsSent}$$

A Javascript script was written and run during the execution of each scenario using the simulation script editor available from Cooja’s tools to compute these defined metrics. The script analyzes the log listener and provides real-time statistics as defined in the written script. The script editor is shown in Figure 15.

### 7.4. Results Analysis and Discussion

The results of the three simulated scenarios (no-attack, buffer reservation attack, countermeasures with buffer reservation attack) are displayed in Figure 16, Figure 17, Figure 18 and Figure 19 and Table 5.

Throughput

As depicted in Figure 18 and Table 5, it can be observed that the throughput is higher in the no-attack scenario, as the buffer reservation attacks cause a 67% decrease in throughput with the 60 s interval. The countermeasure slightly improved the throughput. This slight improvement may be attributed to the server having more time to receive and reassemble other legitimate packets after blocking the attacks. However, throughput improvement is minimal, seeing that the countermeasure merely enables the server to block and stop receiving the attack packets. Therefore, the attack packets are lost and do not increase the throughput. The throughput of the no-attack scenario increased when Node 3 decreased the interval of sending packets, which is expected because, in this scenario, Node 3 sends all the packet’s fragments. The throughput of the attack and attack with countermeasure scenarios are almost identical in the different intervals of sending malicious packets. This is because the throughput measures the amount of received data over time and does not account for the amount of sent data. The stability of the throughput across different attack intervals in this topology indicates that even though the server increasingly waits and times out, it can still receive the same amount of data (from Node 4) as the attack interval changes. Perhaps with a larger WSN, the attack interval would have more effect on the throughput, as the server would struggle to keep up with receiving the legitimate packets while wasting time waiting for the malicious incomplete packets.

Packet Delivery Ratio

As depicted in Figure 17 and Table 5, evidently, the packet delivery ratio is the highest in the no-attack scenario, with a 100% packet delivery ratio. In the attack scenario, we can see that the packet delivery ratio significantly drops from 100% to 35% with an attack interval of 60s. This is because the attack packets are incomplete and are not fully received by the server. Furthermore, as the server is busy waiting for incomplete packets from the attacker, some legitimate packets may be lost. The countermeasure slightly improves the packet delivery ratio. However, the improvement is minimal because most lost packets are blocked malicious packets.

Packet Drop Ratio

As depicted in Figure 16, and Table 5, there were 0% dropped data packets in the normal network scenario. In the attack scenario, the packet drop ratio increases from 0% to more than 50%, as more than half of the sent data packets are dropped due to the attack. Furthermore, it can be observed that the drop ratio increases as the attacker node (Node 3) shortens the interval of sending malicious packets. This is because more malicious packets are sent by the attacker and consequently dropped by the victim as it waits for the remaining fragments. Moreover, it is evident that the countermeasure dramatically decreases the packet drop ratio by more than 50%. This is because the victim node can receive more packets after the attacker node is blocked. Thus, as the attacker’s reputation score declines and ultimately reaches a point where the victim node blocks its malicious FRAG1s, the target node no longer receives the FRAG1s and does not drop the malicious packets. This reflects the success of the countermeasure in mitigating the attack. As the results clearly show, the victim node no longer receives the malicious FARG1s and therefore does not waste time waiting for the remaining FRAGNs.

Average Energy Consumption

As depicted in Figure 19 and Table 5, the average energy consumption of the network in the no-attack normal mode is more than in the other two scenarios because all the packets are legitimate in this scenario. Therefore, all packet fragments are sent and delivered to their destination. Hence, more energy is consumed as more packets are delivered in this case. Most importantly, we can see that the countermeasure decreases the average energy consumption in each attack interval during the attack. The average power consumption decrease is around 64 mJ compared to the second scenario (attack without the countermeasure). This is because the victim node without the countermeasure receives the malicious FRAG1 packets and consumes more energy waiting for the remaining FARGNs until the timeout is reached. However, with the countermeasure, the target node, according to the attacker’s reputation score and repetitive malicious behavior, actively blocks the attacker’s FRAG1 packets and does not wait for the remaining fragments. Therefore, the target node consumes less energy than the attack scenario without the countermeasure. It is worth noting that the decrease in energy consumption is relatively small.

In summary, the conducted experiments reveal the negative impact of the buffer reservation attack on the WSN, particularly the decline of the packet delivery ratio, the throughput, and the packet drop ratio. Moreover, the simulations reflected the effectiveness of the implemented reputation-scoring countermeasure in mitigating the attacks. The countermeasure effectively improved the packet drop ratio and the average energy consumption of the network.

Sensors in WSNs are energy-constrained devices with limited power supply; hence, power consumption is one of the major performance metrics that needs to be investigated and used to show the effectiveness of our proposed countermeasure. We examined the correlation between average energy consumption with (a) packet delivery ratio and (b) throughput as illustrated in Figure 20 and Figure 21, respectively. Figure 20 shows that our proposed countermeasure achieves similar throughput as the no-attack normal mode while consuming less energy. Similar results are found in Figure 21, where our proposed countermeasure achieves similar packet drop ratio as the no-attack normal mode while consuming less energy.

### 7.5. Comparison with other Buffer Reservation Countermeasures

Our implemented countermeasure has advantages over other, more complex buffer reservation countermeasures. Table 6 highlights some of these advantages. The most prominent advantage of our countermeasure is energy efficiency. Previous reputation-scoring countermeasures [3,14] cause an increase in energy consumption due to the complexity of their countermeasures. The BRAIN countermeasure [14] causes a 13.5% increase in energy consumption compared to the original Contiki system. The SecuPAN model [15] leads to a 33% increase in power consumption. On the other hand, the model implemented in this paper does not require additional energy and even helps reduce energy consumption slightly in the attack scenario, as seen in Figure 22.

Regarding PDR, our countermeasure causes an average decrease of PDR across different packet sending intervals of 53.6%, which is 36% better than the improvement in PDR reported in BRAIN [14]. Unfortunately, [15] did not report the improvement in PDR with their proposed countermeasure SecuPAN; hence, we cannot compare our method to [15].

### 7.6. Limitations

In this section, we mention some limitations of our study. One flaw of the implemented countermeasure is that the attacker is permanently blocked when its reputation score drops to a certain threshold. However, a mechanism can be defined to re-evaluate a blocked node if its behavior improves and the receiver gradually regains trust. This is because a node may experience a malfunction that repeatedly disallows the node from sending complete packets and, therefore, the receiver node may permanently block the sender node. Hence, a measure should be defined to accommodate such unintentional harmful behavior. Another mechanism can be defined to provide some incentives to encourage and reward trustworthy nodes.

We also note that because of the limited time to conduct simulations, we did not experiment with other network settings, such as experimenting with different numbers of nodes, having more than one malicious node, and sending messages with varying sizes across the network. It would have been interesting to analyze the implications of these varying settings on the network performance and the effectiveness of the implemented countermeasure.

## 8. Conclusions and Future Work

Wireless sensor networks suffer from several security vulnerabilities. The 6LoWPAN adaptation layer is a major concern, particularly within the packet fragmentation mechanism. This is due to the lack of fragment authentication and integrity verification at this layer. This article describes potential security attacks related to the 6LoWPAN fragmentation attacks, such as fragment replay, fragment alteration, and buffer reservation. Considering that buffer reservation attacks threaten the network service availability and inflict the most harm, this article focuses on this attack in particular. Network analysis was conducted to assess and understand the buffer reservation attack’s impact on the network. The buffer reservation attack was simulated using Contiki Cooja. Several evaluation metrics were defined and analyzed to compare the network’s performance with and without the attack. The attack caused a severe decline in network packet delivery, throughput, and packet drop ratios. Moreover, it was observed in the literature that a reputation-scoring scheme is often used to counter this attack. Therefore, a reputation-scoring scheme was implemented and assessed in this paper. The simulations reflected the effectiveness of the implemented reputation-scoring countermeasure in mitigating attacks. The countermeasure effectively improves the packet drop ratio and the average energy consumption of the network. Moreover, future work may include simulating several other fragmentation attacks, such as fragment duplication, alteration, and spoofing, and improving the countermeasure to accommodate them. Finally, a comparative analysis may be conducted to compare each attack’s effects empirically.

## Figures and Tables

**Figure 1 sensors-22-09825-f001:**
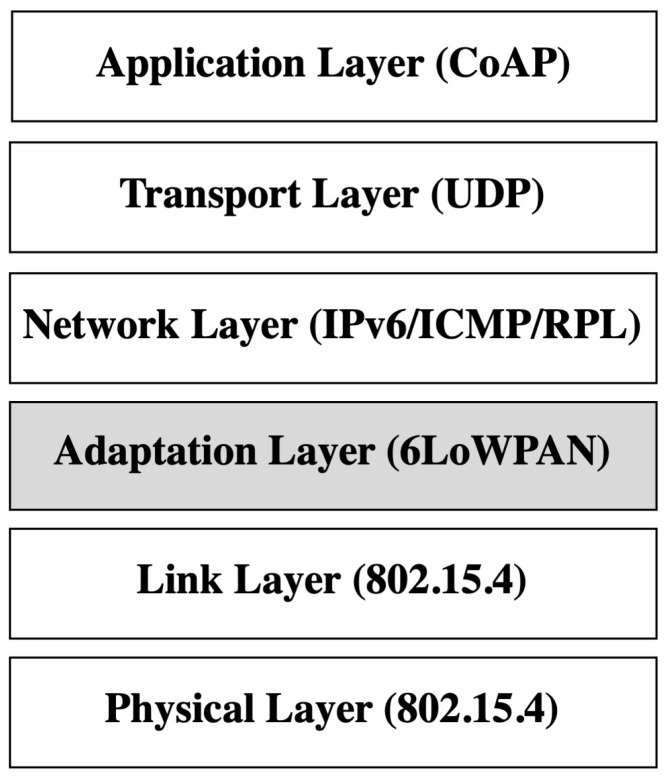
6LowPAN layers—adapted from [4].

**Figure 2 sensors-22-09825-f002:**
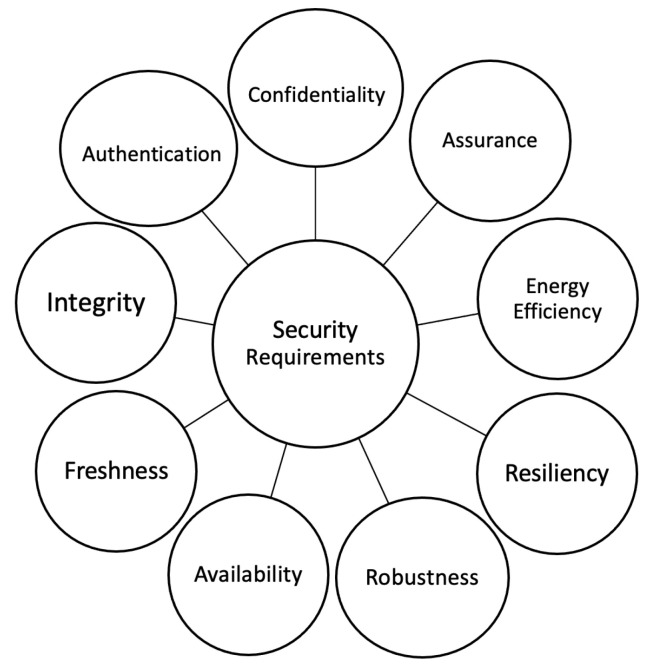
6LoWPAN security requirements.

**Figure 3 sensors-22-09825-f003:**
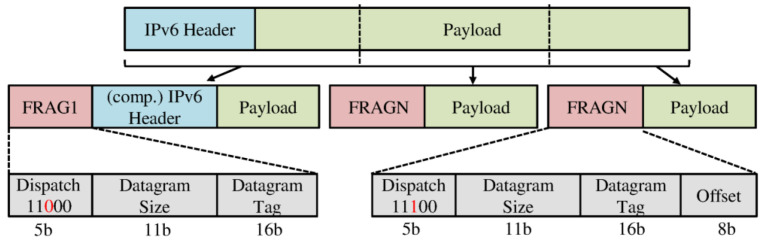
Packet fragmentation—adapted from [3].

**Figure 4 sensors-22-09825-f004:**
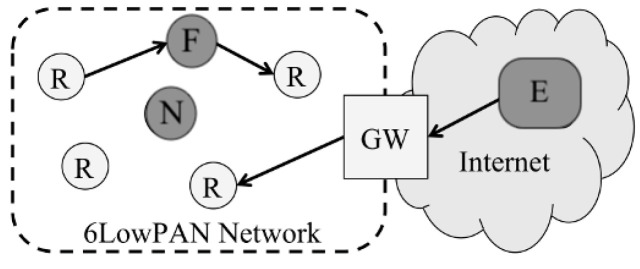
Network scenario.

**Figure 5 sensors-22-09825-f005:**
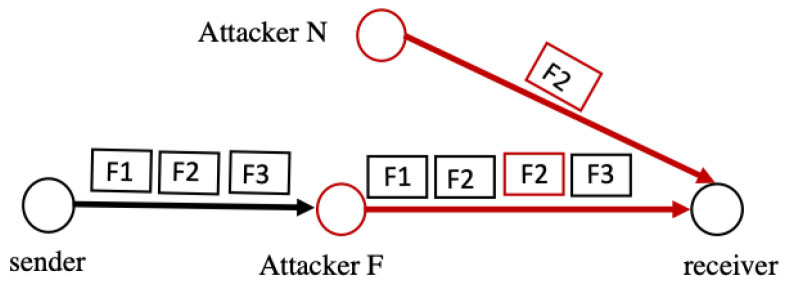
Duplication attack.

**Figure 6 sensors-22-09825-f006:**
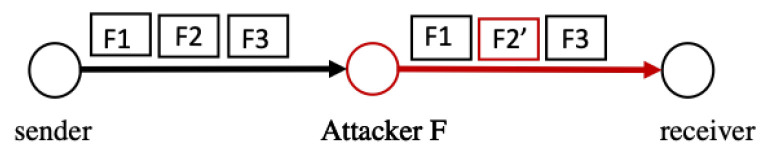
Alteration attack.

**Figure 7 sensors-22-09825-f007:**
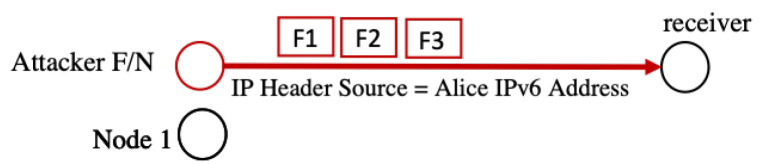
Spoofing attack.

**Figure 8 sensors-22-09825-f008:**
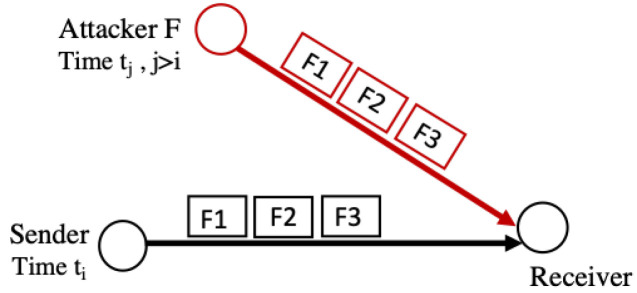
Replay attack.

**Figure 9 sensors-22-09825-f009:**
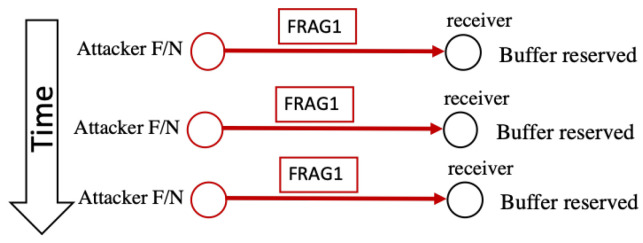
Buffer reservation attack.

**Figure 10 sensors-22-09825-f010:**
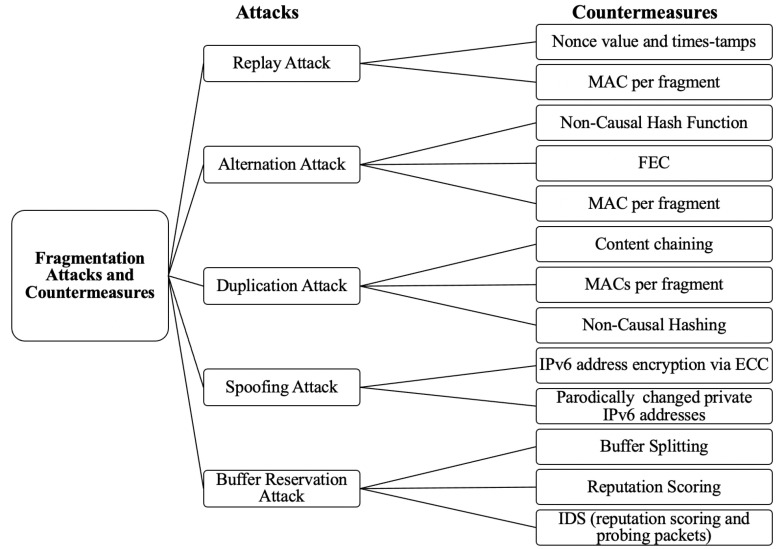
Taxonomy of fragmentation attacks and countermeasures.

**Figure 11 sensors-22-09825-f011:**
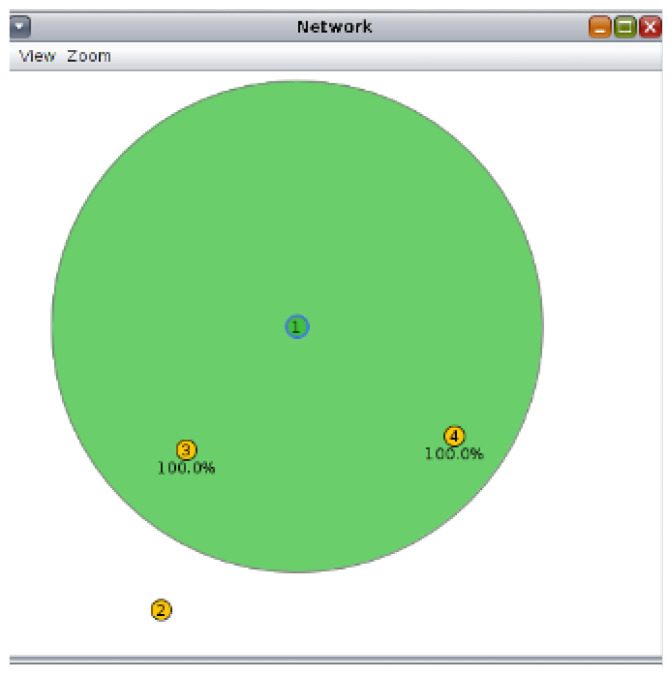
Network topology.

**Figure 12 sensors-22-09825-f012:**
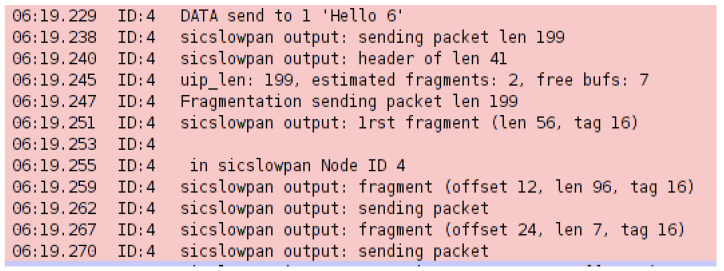
Log listener—normal packet fragmentation.

**Figure 13 sensors-22-09825-f013:**
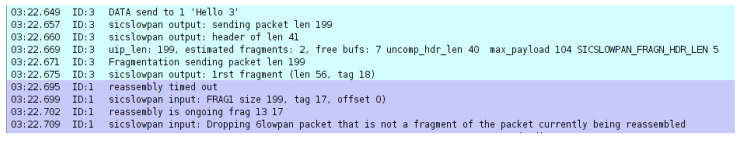
Log listener—packet fragmentation with attack.

**Figure 14 sensors-22-09825-f014:**
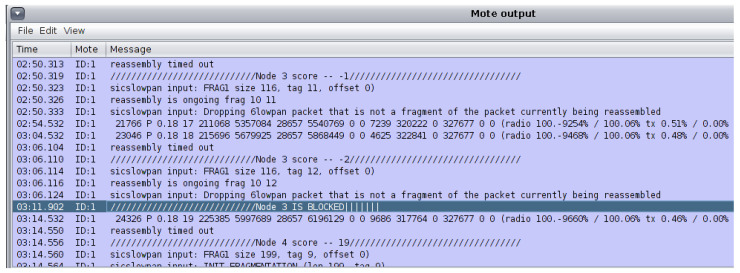
Log listener—server blocking Node 3 attacks using countermeasure.

**Figure 15 sensors-22-09825-f015:**
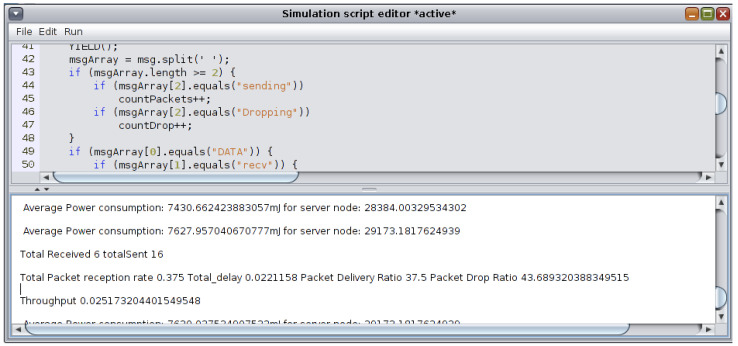
Cooja simulation script editor.

**Figure 16 sensors-22-09825-f016:**
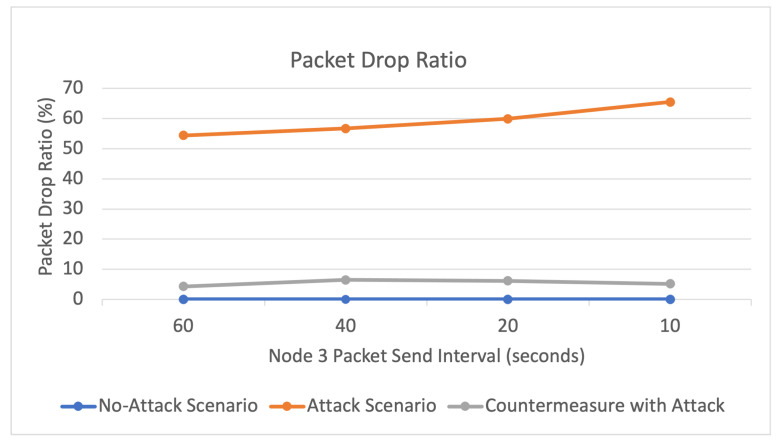
Packet drop ratio.

**Figure 17 sensors-22-09825-f017:**
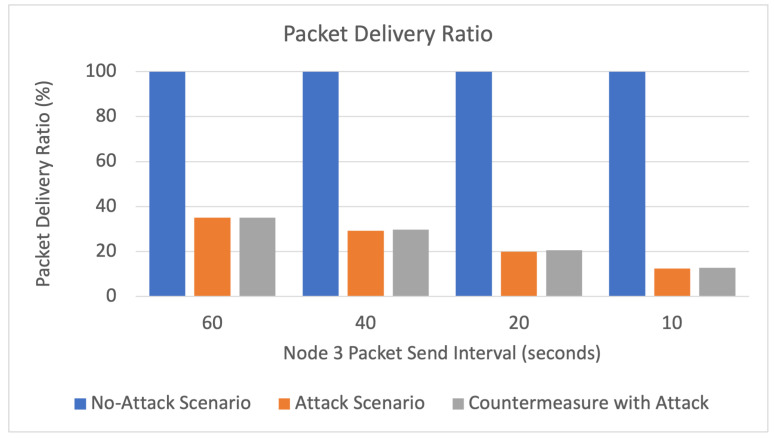
Packet delivery ratio.

**Figure 18 sensors-22-09825-f018:**
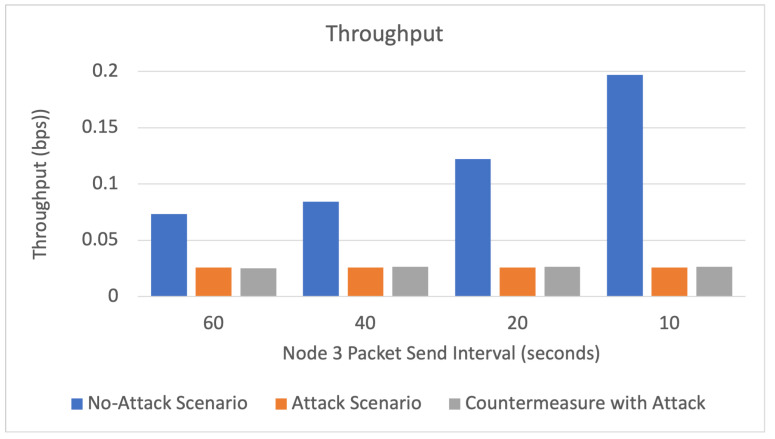
Throughput.

**Figure 19 sensors-22-09825-f019:**
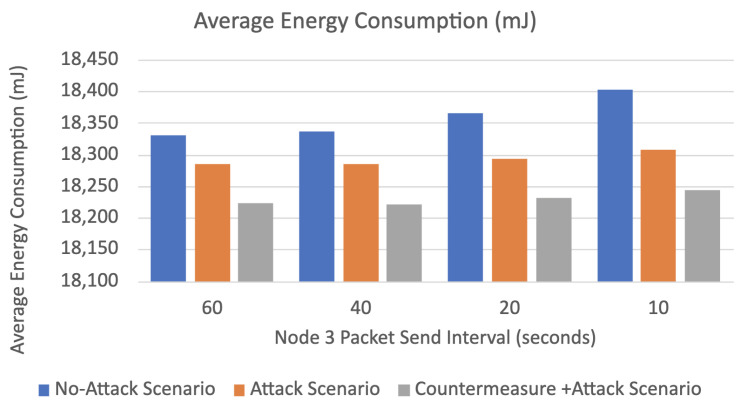
Average energy consumption.

**Figure 20 sensors-22-09825-f020:**
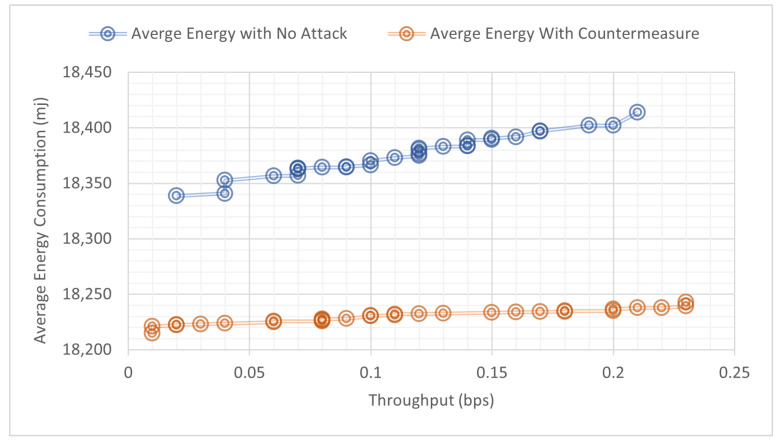
Correlation of throughput and average energy consumption.

**Figure 21 sensors-22-09825-f021:**
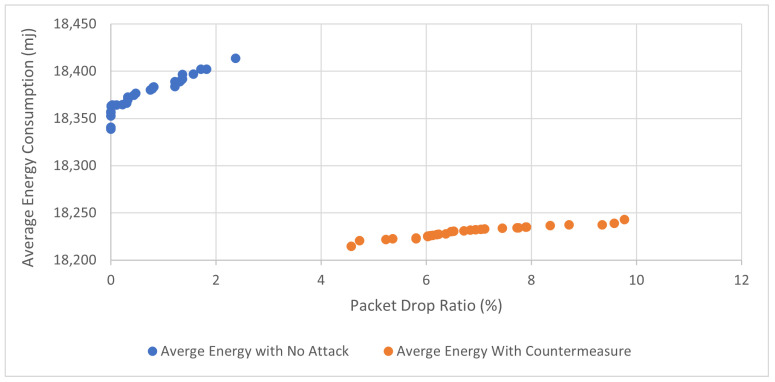
Correlation of packet delivery ratio and average energy consumption.

**Figure 22 sensors-22-09825-f022:**
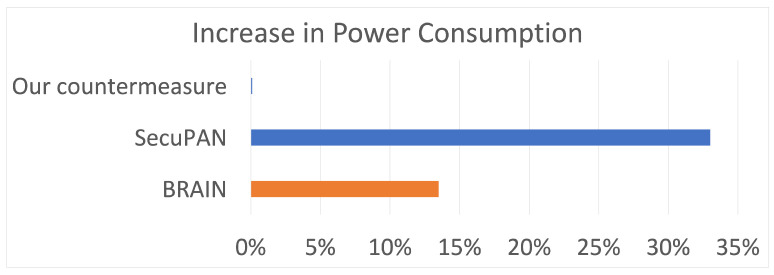
Increase in power consumption compared to normal Contiki.

**Table 1 sensors-22-09825-t001:** Literature review summary.

Ref., Year	Attacks	Proposed Solution	Tool	Comments
[24], 2008	Replay	Nonce, timestamps	NA	Scheme not implemented and tested
[16], 2013	Duplication, buffer reservation	Content chaining and buffer splitting	Contiki	Requires that the fragments be sent in order. If one fragment is lost, cannot verify hash chain.
[28], 2017	Spoofing	Simulation and analysis of spoofing attack	Contiki, Cooja	Analysis provided, but no countermeasure proposed.
[3], 2018	Alteration, duplication, replay, spoofing, buffer reservation	SecuPAN scheme, MACs for each fragment, and reputation-based buffer management	Contiki	Provides less energy consumption. needs only to resend fabricated fragments, not entire packet, but has long run time, especially at receiver end.
[15], 2019	Duplication, alteration	Non-causal hash function scheme (NCHFS)	MATLAB	Proposed scheme not explained sufficiently.
[26], 2019	Duplication	A lightweight online/offline signcryption scheme	NA	Scheme not implemented and tested.
[14], 2020	Buffer reservation	Scheme named “BRAIN” based on legitimacy points.	Contiki, Cooja	Attacker may adopt changing behaviors (on-off attacks).
[11], 2020	Alteration	Dynamically enabled FEC	Custom WSN	Does not address other fragmentation vulnerabilities and attacks.
[17], 2020	Spoofing	Private addressing scheme	Contiki, Cooja	Communication overhead from temporary address management.
[38], 2020	Buffer reservation	ArsPAN IDS based on reputation scoring and probing packets	Contiki	Attacker may adopt changing behaviors (on/off attacks).
[36], 2021	Buffer reservation	Use trust values obtained from integrating RPL chained secure mode	Contiki Cooja	Method fails if attacker uses victims’ data link address instead of IPv6.

**Table 2 sensors-22-09825-t002:** Comparative analysis of fragmentation attack countermeasures.

Ref.	Year	Replay	Alteration	Duplication	Spoofing	Buffer Reserv.
[24]	2008	✓	χ	χ	χ	χ
[16]	2013	χ	χ	✓	χ	✓
[3]	2018	✓	✓	✓	✓	✓
[15]	2019	χ	✓	✓	χ	χ
[26]	2019	χ	χ	✓	χ	χ
[14]	2020	χ	χ	χ	χ	✓
[17]	2020	χ	χ	χ	χ	✓
[11]	2020	χ	✓	χ	χ	χ
[36]	2021	χ	χ	χ	χ	✓
[38]	2020	χ	χ	χ	χ	✓

**Table 3 sensors-22-09825-t003:** Experiment set-up.

Parameter Name	Value
Operating system	Contiki 3.0
Simulation tool	Cooja
Node type	Sky Mote
Radio environment	UDGM
Routing protocol	RPL
Reassembly buffer timeout	Default (20 s)
Number of nodes	4
Transmitted message length	199B
Energy consumption report rate	10 s
Simulation time	15 min

**Table 4 sensors-22-09825-t004:** Sky Mote power specifications.

Attribute	Value
Voltage	3.6 V
Current transmit	19.5 mA
Current receive	21.8 mA
Current CPU	1800 mA
Current idle	54.5 mA

**Table 5 sensors-22-09825-t005:** Results of the simulation (no-attack, buffer reservation attack, countermeasure with buffer reservation attack).

Scenario	Interval	Throughput	Packet Delivery Ratio	Packet Drop Ratio	Average Energy	Node Energy
No attack	60	0.07	100	0	18,330.65	70,170.39
	40	0.08	100	0	18,337.02	70,177.25
	20	0.12	100	0	18,365.38	70,201.17
	10	0.20702	100	0	18,403.15	70,244.73
	Mean	0.11	96.63	0.79	18,380.48	70,199.57
	SD	0.04	1.98	1.34	23.61	27.43
Attack	60	0.025	35	54.40	18,284.78	70,162.16
	40	0.025	29.17	56.72	18,285.73	70,162.60
	20	0.02570	20	59.90	18,292.65	70,165.28
	10	0.02	12.50	65.53	18,307.14	70,171.16
	Mean	0.02	24.23	56.73	18,308.42	70,166.03
	SD	0	8.56	4.04	8.99	3.01
Attack with countermeasure	60	0.02640	35	4.27	18,222.28	70,146.80
	40	0.02645	29.79	6.43	18,220.97	70,147.13
	20	0.02640	20.59	6.14	18,230.22	70,148.98
	10	0.26450	12.73	5.15	18,242.69	70,151.30
	Mean	0.02	24.52	6.62	18,229.04	70,148.66
	SD	0	8.54	1.24	8.64	1.69

**Table 6 sensors-22-09825-t006:** Comparison with other countermeasures.

Countermeasure	Improvement in PDR	Increase in Power Consumption
BRAIN [14]	17.25%	13.5%
SecuPAN [3]	NA	33%
**Our proposed countermeasure**	**53.64**%	**0%**

## Data Availability

Not applicable.

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
