# Peer review of "Fragmentation Attacks and Countermeasures on 6LoWPAN Internet of Things Networks: Survey and Simulation"

_sensors, 2022, doi:10.3390/s22249825_

Round 1

Reviewer 1 Report

Fragmentation Attacks and Countermeasures on 6LoWPAN Internet of Things Networks: Survey and Simulation

This article provides a survey of fragmentation attacks and available countermeasures. Furthermore, the Buffer Reservation attack, one of the most harmful fragmentation attacks that may cause DoS, is studied and simulated in detail. Additionally, a countermeasure for this attack is implemented based on a reputation scoring scheme. The conducted experiments show the harmful effect of the Buffer Reservation attack and the effectiveness of the implemented reputation scoring countermeasure.

However, I have only a few minor comments after reading the manuscript.

There are some typos, article mistakes, and grammatical errors in the manuscript. For instance, in the introduction “6LoWPAN vulnerabilities are inherited 44 from the Internet Protocol (IP) and the resource-constrained nature pf the devices”. Please review and revise the whole manuscript to remove typos.

The quality of figure 1 is poor. Please enhance the quality of the figure in the revised version to help the reader.

Author Response

First of all, we would like to thank the reviewers for their time in reading our manuscript and providing valuable comments to improve the quality of our manuscript.  Here, we are writing this to inform you that we have addressed the issues raised by all reviewers. These changes have been incorporated in the revised manuscript. Please see the attached PDF document. Thank you again.

Reviewer 2 Report

This paper provides a survey of fragmentation attacks and available countermeasures. Overall, the topic is interesting, and the proposed solution provides interesting results. The following comments are recommended to improve the quality of the paper.

(1)  In Section 1, the introduction does not read smoothly. The storyline between the existing work and the methodology of this paper has not been sorted out.

(2)  In Section 1, the contributions of this article are better listed separately.

(3)  In Section 4, your proposed method to handle the attack should be specifically summarized.

(4)  In Section 7, the advantages of the proposed method should be highlighted by comparing it with existing solutions.

(5)  The survey of this paper requires further improvement. On the detection of attacks, “Statistical Approach to Detection of Attacks for Stochastic Cyber-Physical Systems, IEEE trans on Automatic Control”. On the analysis for the system under attacks, “The Vulnerability of Cyber-Physical System Under Stealthy Attacks, IEEE trans on Automatic Control”. I believe that theses works may help to improve the quality of this paper.

(6)  The limitations of this study are not found. Also, A brief description of future work should be given in the conclusion.

(7)  To help readers quickly understand, adding some remarks is suggested.

Author Response

(The authors gave the same response as above.)

Reviewer 3 Report

-       Summary  

This paper displays a survey of fragmentation attacks and available countermeasures. It simulated a fragmentation attack called the Buffer Reservation attack that leads to Denial OF Service (DOS). They used and analyzed several evaluation metrics to compare the network’s performance with and without the attack.

-       Strengths

·      The title is appropriate as it is short, clear, and understandable.

·      They commented on all figures in a clear and detailed way that helps in understanding it.

·      They provided an enough overview of all terminologies that are mentioned in this manuscript.

-       Weaknesses

·      They must expand the survey by adding more related works from 2021 -2022.

·      They must demonstrate more experiments to justify the effectiveness of their work. 

·      The main contribution in the manuscript must be specified clearly in the introduction section.  Please, highlight your contributions in the introduction with more details.

·      The motivation is not clear. Please explain it clearly.

·      They must apply some statistics to all of the simulation results to show their significance.

·      Finally, I believe that this paper is good but it suffers from some problems that must be solved.  Good revision is needed before it can be considered for publication in this journal.

Author Response

(The authors gave the same response as above.)
